# Comparing Fully Deep Convolutional Neural Networks for Land Cover Classification with High-Spatial-Resolution Gaofen-2 Images

**Zemin Han** [1], **Yuanyong Dian** [1,2,3,*] , **Hao Xia** [4], **Jingjing Zhou** [1,2], **Yongfeng Jian** [1], **Chonghuai Yao** [1,2], **Xiong Wang** [1] **and Yuan Li** [1]

[1] College of Horticulture and Forestry Sciences, Huazhong Agricultural University, Wuhan 430070, China; HZM@webmail.hzau.edu.cn (Z.H.); hupodingxiangyu@mail.hzau.edu.cn (J.Z.); JYongFeng@webmail.hzau.edu.cn (Y.J.); yao_chonghuai@mail.hzau.edu.cn (C.Y.); wangxiong@webmail.hzau.edu.cn (X.W.); liyuan994@webmail.hzau.edu.cn (Y.L.)

[2] Hubei Engineering Technology Research Centre for Forestry Information, Huazhong Agricultural University, Wuhan 430070, China

[3] Key Laboratory of Urban Agriculture in Central China, Ministry of Agriculture, Wuhan 430070, China

[4] Aerospace Information Research Institute, Chinese Academy of Science, Beijing 100101, China; xiahao@radi.ac.cn

* Correspondence: dianyuanyong@mail.hzau.edu.cn; Tel.: +86-027-87282010

**Abstract:** Land cover is an important variable of the terrestrial ecosystem that provides information for natural resources management, urban sprawl detection, and environment research. To classify land cover with high-spatial-resolution multispectral remote sensing imagery is a difficult problem due to heterogeneous spectral values of the same object on the ground. Fully convolutional networks (FCNs) are a state-of-the-art method that has been increasingly used in image segmentation and classification. However, a systematic quantitative comparison of FCNs on high-spatial-multispectral remote imagery was not yet performed. In this paper, we adopted the three FCNs (FCN-8s, Segnet, and Unet) for Gaofen-2 (GF2) satellite imagery classification. Two scenes of GF2 with a total of 3329 polygon samples were used in the study area and a systematic quantitative comparison of FCNs was conducted with red, green, blue (RGB) and RGB+near infrared (NIR) inputs for GF2 satellite imagery. The results showed that: (1) The FCN methods perform well in land cover classification with GF2 imagery, and yet, different FCNs architectures exhibited different results in mapping accuracy. The FCN-8s model performed best among the Segnet and Unet architectures due to the multiscale feature channels in the upsampling stage. Averaged across the models, the overall accuracy (*OA*) and *Kappa* coefficient (*Kappa*) were 5% and 0.06 higher, respectively, in FCN-8s when compared with the other two models. (2) High-spatial-resolution remote sensing imagery with RGB+NIR bands performed better than RGB input at mapping land cover, and yet the advantage was limited; the *OA* and *Kappa* only increased an average of 0.4% and 0.01 in the RGB+NIR bands. (3) The GF2 imagery provided an encouraging result in estimating land cover based on the FCN-8s method, which can be exploited for large-scale land cover mapping in the future.

**Keywords:** deep learning; classification; land cover; full convolutional network; Unet; Segnet; FCN

---

## 1. Introduction

Land cover is an important variable of the terrestrial ecosystem and provides information for natural resources management, urban sprawl detection, and environmental research [1]. Fine resolution land cover maps and their change over time can offer more information to manage the Earth. However,

mapping the land cover map in a local or global scale was a problem for ecologists due to a lack of large regional scale data [2].

Remote sensing has long been recognized as a good way to solve this problem. Land cover classification with remote sensing imagery is a widely spread research topic in the world [3–7]. A variety of land cover products are generated from different sensors at global or regional scales, such as the moderate resolution imaging spectroradiometer (MODIS) Land Cover Type Product (MCD12Q1) [6], Globe Land Cover 30-2010 (GLC30) derived from Landsat series imagery [1], and Finer Resolution Observation and Monitoring of Global Land Cover (FROM-GLC) derived from Sentinel imagery [8,9].

The number of land cover types classified and the accuracy of the classification products are often limited due to its coarse spatial resolution. MCD12Q1 provides global land cover at 500 m spatial resolution at annual time step for 17 types of land cover, including 11 types of natural vegetation, three types of developed and disordered land, and three types of non-vegetation land. GLC30 provides a 30 m resolution land cover product on a global scale with the baseline year of 2010 and contains 10 land cover types. FROM-GLC products provide 30 m and 10 m resolution land cover productions, which contain 10 class types. The classification schemes are sufficient on a global or regional scale; however, they are too coarse to distinguish different forest types or crop types in the local area, and, due to the heterogeneity of landscapes, the overall accuracy of these products is not over 80% [1,2,9].

With the development of remote sensing technology, a large number of high-resolution images (<10 m) can be collected in a short period, such as the Satellite for observation of Earth (SPOT), Worldview, and Gaofen-2 (GF2) images [10–12]. The GF2 satellite, which launched in 2014 and is configured with two panchromatic and multispectral charge coupled device (CCD) camera sensors, is a Chinese high resolution optical satellite and can achieve a spatial resolution of 1 m in the panchromatic mode and a resolution of 4 m in four spectral bands in the multispectral mode with a swath width of 45 km [13]. GF2 data have a high frequency revisit time of five days. The high spatial resolution and high frequency revisit time, as well as the wide coverage ability of GF2 data, make them highly suitable data sources for land cover change detection. Therefore, it is necessary to develop a classification algorithm for GF2. Recently, researchers focused on land cover map with the fine spatial resolution satellite imagery [7,10,12,14–17].

Generally, the methods of land cover classification can be cataloged into two ways, non-deep methods and deep learning methods [18,19]. In a non-deep way, the primary methods rely on single-pixel based and object-based methods. The single-pixel-based algorithms mainly focus on the spectral signature and are used for moderate or coarse spatial resolution imagery. This may lead to misclassification of pixels due to the similar spectral signatures of different objects. Object-based methods are primarily used for high-spatial-resolution imagery, which consider the spatial patterns combined with the spectral signatures in a classification algorithm.

All the features extracted from the object-based method were fed to classifiers, such as support vector machine (SVM) or random forest (RF), to classify the land cover types. These approaches in high-spatial-resolution imagery demonstrated good achievements in land cover classification [10,20]. However, the parameters (the minimum size of an object) defined in the object-based method were also a limitation which would reject areas smaller than the minimum size. The object-based way is also a low-level feature derived method, where the classification accuracy depends on the object segment results.

Deep learning (DL) algorithms which use deep neural networks (DNN) were quickly developed in recent years and achieved significant success at many image analysis tasks, especially in scene classification, object detection, and land cover classification [18,19,21]. The underlying advantage of DL is that it can automatically extract features from data using an optimization loop, which can improve the efficiency when compared with manual feature extraction methods.

Convolutional neural networks (CNNs) are the most well-known deep learning methods for feature extraction, particularly for 2-D spatial features, by using a hierarchy of convolutional filters with nonlinear transformations. CNNs are effective at image classification especially for scene classification. The patch-based CNNs are commonly used in pixel-level image classification. Yet, patch-based CNNs are inefficient due to the densely overlapped patch size in processing the whole remote sensing imagery.

Recently, fully convolutional networks (FCNs) were proposed to overcome the drawbacks in patch-based CNNs. FCNs are an innovative method to address pixel-based classification in an end-to-end approach, by proposing a downsample-then-upsample architecture in encoder-decoders. Comparing CNN and FCN, researchers found that both have encoder layers used to extract features from the image with the downsample strategy, while the FCN contains decoder layers that use skip connections to fuse the information from encoder layers in the network and upsampled the image to the original size.

Several FCN architectures have been proposed in recent years. FCN-8s, FCN-16s, FCN-32s, Segnet, and Unet are common ones used in pixel-based classification [22,23]. These architectures are similar at the encoder stage but different at the decoder stage. Despite many proposed models that were reported in previous studies, a systematic quantitative comparison of these FCNs has not been well performed. The majority of pre-trained networks are currently limited to red, green, and blue (RGB) input data, and few have used spectral satellite image as an input, in particular for the near infrared (NIR) band.

The objectives of this paper were: (1) to evaluate the potential of FCNs on GF2 imagery in land cover classification; (2) to conduct a systematic quantitative comparison between different FCNs; (3) to compare the classification accuracy when feeding RGB and RGB+NIR as input for GF2 satellite imagery.

## 2. Materials and Methods

### 2.1. Study Area and Data

The study was conducted on Xinxiang city in Henan province in central China (35°27′11″ N–35°45′43″ N, 113°27′32″ E–113°39′25″ E). The characteristics of land cover in this area are traditional forest landscape and farmland mosaic with villages in central China, which represent a good test for classification.

We used two scenes of GF2 images in the study area (Figure 1), which were acquired on 1 April, 2017. Training samples were only selected from S2, and the testing samples were selected both from S1 and S2 in order to test the training transferring ability. The image size of the two scenes was 7300 × 6908 pixels. Both scenes were preprocessed with a quick atmospheric correction method and geometrical rectification in ENVI 5.3 (Exelis Visual Information Solutions, Boulder, CO, USA).

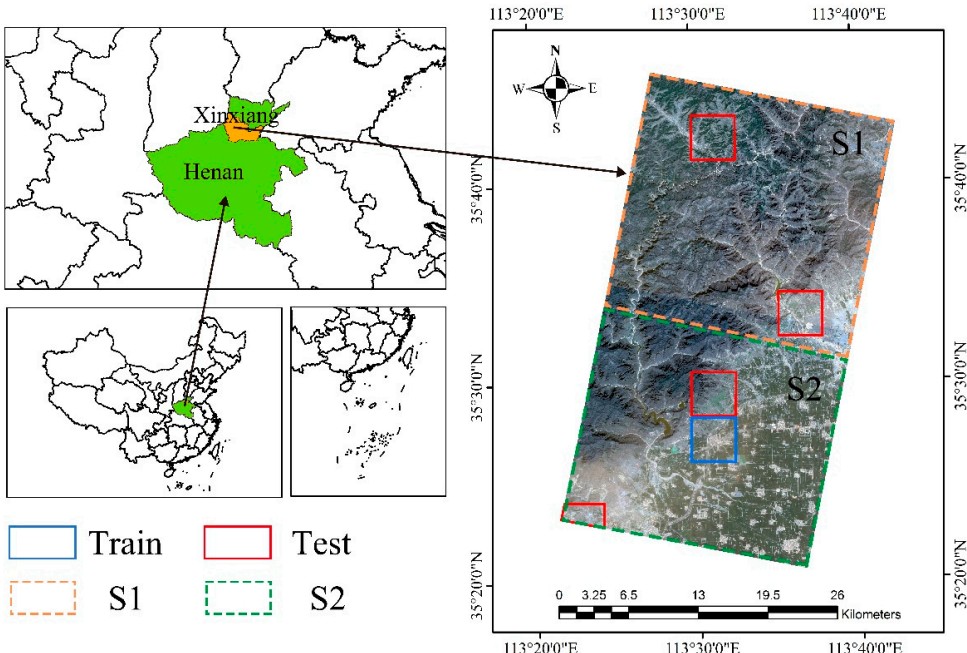

**Figure 1.** Location of the study area (leaf) and two scenes (S1 and S2) of Gaofen-2 (GF2). Train and test sample areas are labeled with a different color (blue for training and red for testing).

## 2.2. Land Cover Classes

There are thirteen dominant land cover classes in the research area including Cereal (CL), Grass Land (GL), Poplar Land (PL), Pathway in Farmland (PF), Impervious Road (IR), Sparse Woods (SW), Dense Woods (DW), Water Bodies (WB), Waterless Channel (WC), Water Conservancy Facilities (WF), Construction Land (CD), Greenhouse (GH) and Bare Land (BL) (Table 1).

**Table 1.** Land cover description and the number of available polygons and pixels for training and testing for each class.

| Land Cover Name | Code | Description | Polygon | Train Pixels | Test Pixels |
|---|---|---|---|---|---|
| Cereal Land | CL | Including rice, wheat, millet, soybeans, etc., mainly plant seeds and fruits | 496 | 907,785 | 226,946 |
| Grass Land | GL | Grassland is the land where herbs and shrubs are grown | 660 | 604,179 | 151,045 |
| Poplar Land | PL | Growing poplar | 38 | 18,622 | 4655 |
| Pathway in Farmland | PY | The road for people to walk between the fields | 268 | 48,829 | 12,207 |
| Impervious Road | IR | A road for people to drive through, a passage between the two places | 443 | 83,094 | 20,773 |
| Sparse Woods | SW | Forest land with a canopy cover less than 20% | 198 | 223,582 | 55,895 |
| Dense Woods | DW | Forest land with dense trees and canopy cover greater than 20% | 689 | 942,067 | 235,517 |
| Water Bodies | WB | The collection of water is an important part of the surface water ring | 26 | 14,213 | 3553 |
| Waterless Channel | WC | Refers to the waterway that can be navigable | 44 | 59,933 | 14,983 |
| Water Conservancy Facilities | WF | Land for reservoirs and hydraulic structures | 22 | 20,693 | 5173 |
| Construction Land | CD | This refers to the land where buildings and structures are built | 251 | 229,085 | 57,271 |
| Greenhouse | GH | A facility that can be used to grow plants by transmitting light and maintaining warmth | 46 | 14,641 | 3660 |
| Bare Land | BL | No exposed ground for plant growth | 148 | 114,279 | 28,570 |

A handcraft annotation procedure was carried out to produce reference data. We manually digitized polygons for different land cover classes. In order to acquire more representative samples, we noted as many variations of each class as possible and avoided mixtures of the defined land cover classes. After that, the labeled polygons were rasterized at the same resolution as the input imagery. The detailed information regarding each class is described in Table 1.

## 2.3. Architecture of FCNs

There were several FCNs architectures proposed in recent years. The FCN-8s, FCN-16s, FCN-32s, Segnet, and Unet were common ones used in pixel-based classification. These architectures are similar at the encoder stage but different at the decoder stage (Figure 2).

FCN-8s, FCN-16s, and FCN-32s were the first generation of FCN architectures [24]. These networks combined the final prediction layer with the lower layers with finer strides in the upsample stage. FCN-8s and FCN-16s have two and one skip connections and upsample at stride 8 and 16, respectively, by adding a $2 \times 2$ upsampling layer which sums the semantic information from a deep, coarse layer with appearance information obtained from shallow, fine layers. FCN-32s produced dense pixel-based labeled maps with no skip connection at stride 32 by adding a $2 \times 2$ upsampling layer (Figure 2a).

Segnet is another FCN architecture, which also followed the encoder-decoder paradigm. Segnet uses memorized max-pooling indices which are the location of the maximum feature value in each max-pooling operation during the encoder stage to perform non-linear upsampling during the decoding stage (Figure 2b).

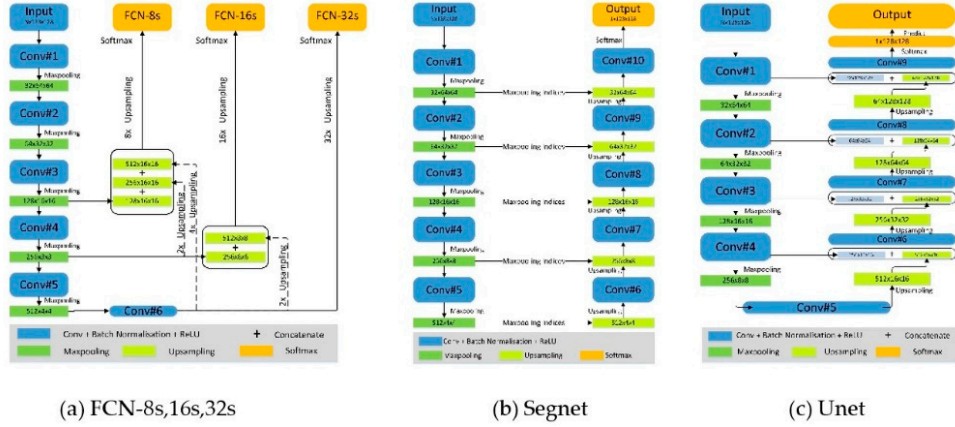

**Figure 2.** The architectures of fully convolutional networks (FCNs) (**a**) FCN-8,16,32s (**b**) Segnet (**c**) Unet. Input means the input image size in the FCNs, where N is the band of input images which equals 3 or 4, and the image patch size is 128 × 128. Conv indicates the convolution layer combined with batch normalization and rectified non-linear function (ReLU). Rectangles labeled with the b × w × h (b, w, and h indicate the feature channel number, image width, and height, respectively) are the output size after max pooling or the upsampling method. The symbol "+" in (**a**), (**c**) indicates different concatenation features at the feature channel dimension. "Maxpooling indices" in (**b**) indicates the locations of the maximum feature value in each pooling window. They are memorized in the max pooling stage and then used in the upsampling stage.

Unet is a modified fully convolutional neural network architecture, which was previously used for the tasks of biomedical image segmentation [23]. The innovation of the Unet architecture was that low-level feature maps in the decoder stage concatenated them with higher-level ones in the equivalent stage in the encoder stage during the upsampling process, which compensated for the lost information in the max polling layer and enabled precise localization. With the concatenating way, many feature channels generated an upsampling part, which allowed propagating the context information to higher resolution layers (Figure 2c).

Notably, all convolutional layers of the network were followed by batch normalization, the rectified non-liner function (ReLU), and dropout, because batch normalization increases the training speed and decreases the network sensitivity to initialization, and the dropout layer helps to reduce the overfit.

### 2.4. Evaluation Metrics

We used the overall accuracy (*OA*), *Kappa* coefficient (*Kappa*), precision accuracy (*PA*), and recall accuracy (*RA*) to quantitatively evaluate the experimental results. Additionally, the combined metric of per class *F*1 was also provided (Equations (1)–(5)).

The overall accuracy quantifies the amount of correctly labeled pixels in all ground truth pixels. The *Kappa* coefficient represents the degree of agreement between the ground truth data and the final labeled map, which compensates for the chance agreement between classes. The *F*1 score is a harmonic mean between precision and recall. It is useful for imbalanced classes.

$$OA = \frac{\sum_k x_{ii}}{N} \tag{1}$$

$$kappa = \frac{N \sum_k x_{ii} - \sum_k x_{i+} \sum_k x_{+i}}{N^2 - \sum_k x_{i+} \sum_k x_{+i}} \tag{2}$$

$$PA = \frac{TP}{TP + FP} \tag{3}$$

$$RA = \frac{TP}{TP + FN} \tag{4}$$

$$F1 = 2 \times \frac{PA \times RA}{PA + RA} \tag{5}$$

Here, $N$ is the total number of pixels in all the ground truth classes, $k$ is the number of classes, $x_{ii}$ is the number of pixels of the $i$th class, which were correctly classified into the $i$th class, $x_{i+}$ is the number of ground truth pixels in the $i$th class and $x_{+i}$ is the sum of the pixels classified into the $i$th class.

For each class, $TP$ is the number of true positive, $FP$ is the number of false positives, and $FN$ is the number of false negatives.

## 3. Results

### 3.1. Classification Results and Visual Assessment

The final classification maps of different methods were obtained for the study sites. To offer a better visualization, four subsets with different band inputs were compared. Figure 3 (RGB bands for input) and Figure 4 (RGB+NIR bands for input) highlight the different land type classification results of the different classifiers.

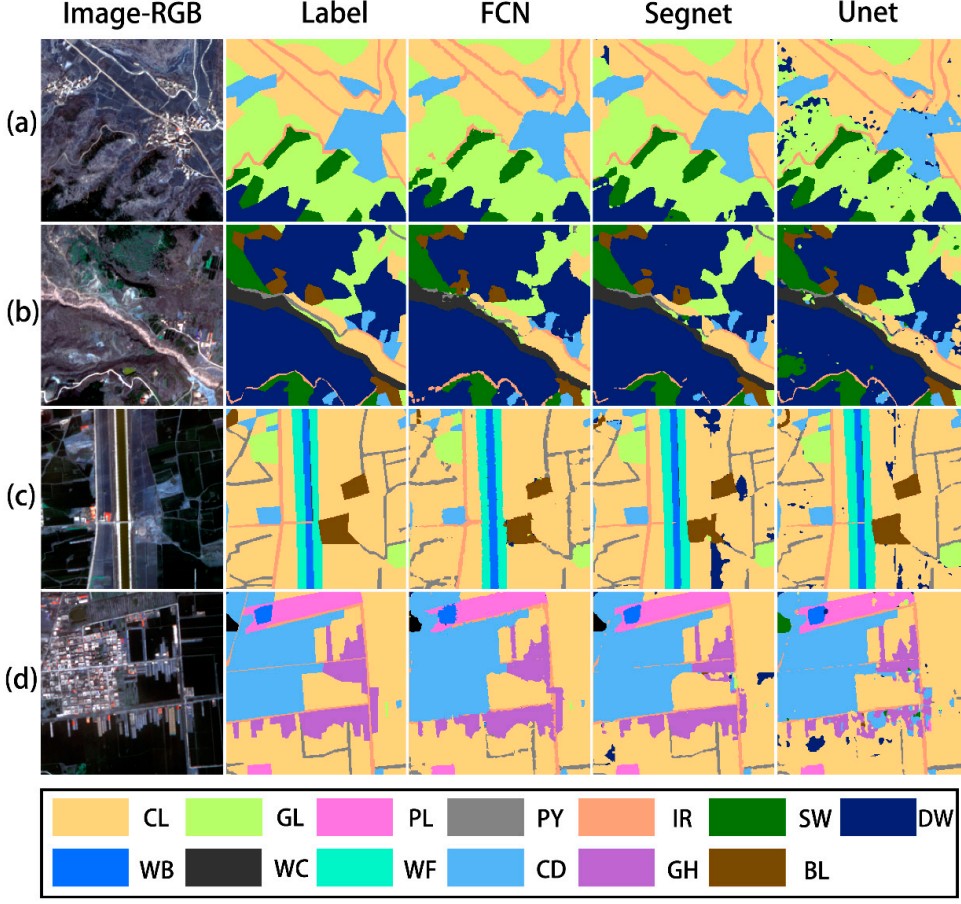

**Figure 3.** Four typical image subsets (**a**–**d**) in the study area with different classification results using only three bands (R+G+B). Columns from the left to right represent the original images (R+G+B bands), label map, FCN-8s classification, Segnet classification, and Unet classification, respectively. CL represents cereal land, GL represents grassland, PL represents poplar land, PY represents pathway in farmland, IR represents the impervious road, SW represents sparse woods, DW represents dense woods, WB represents water bodies, WC represents a waterless channel, WF represents water conservancy facilities, CD represents construction land, GH represents greenhouse, and BL represents bare land.

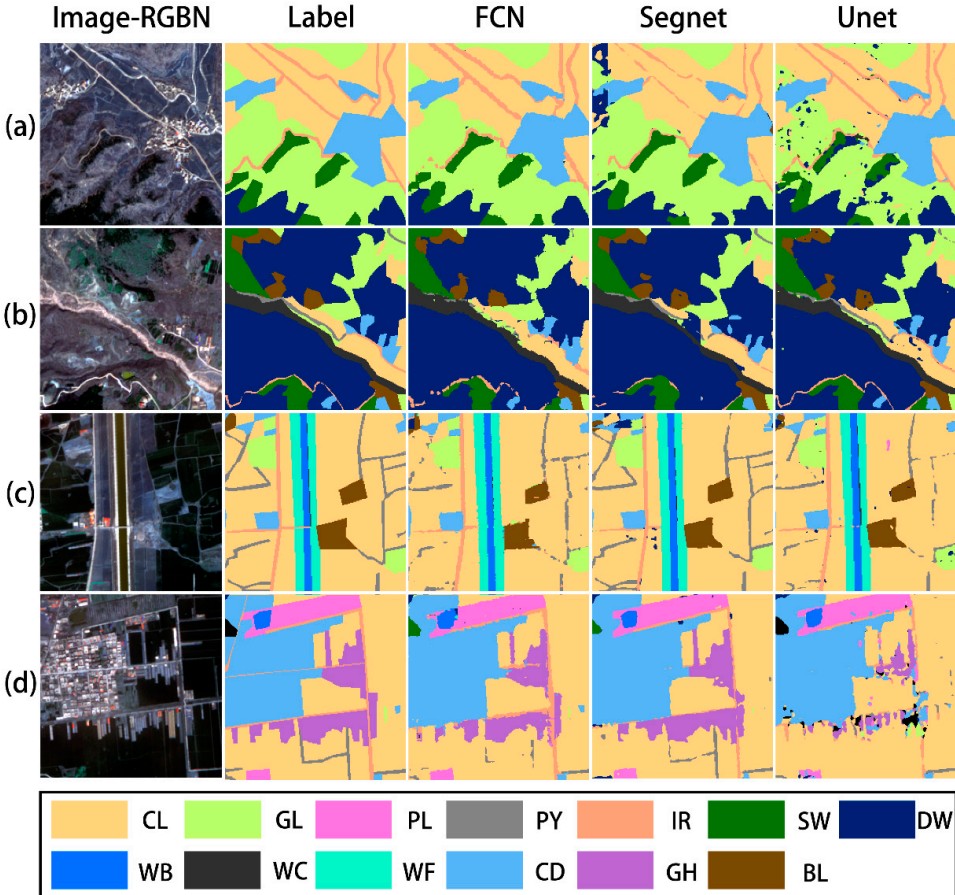

**Figure 4.** Four typical image subsets (**a–d**) in the study area with different classification results using four bands (R+G+B+ near infrared (NIR)). Columns from left to right represent the original images (R+G+B bands), label map, FCN-8s classification, Segnet classification, and Unet classification, respectively. CL represents cereal land, GL represents grassland, PL represents poplar land, PY represents pathway in farmland, IR represents impervious road, SW represents sparse woods, DW represents dense woods, WB represents water bodies, WC represents a waterless channel, WF represents water conservancy facilities, CD represents construction land, GH represents greenhouse, and BL represents bare land.

In contrast to the different input bands, the classification results of FCN in all study sites exhibit smoothed visual effects with the least speckle noise as shown in Figures 3 and 4. We could not find obvious differences in the classification results from the visual assessment. Additionally, the 13 classes, whether they were large or small patches, were accurately distinguished in both three band or four band inputs. The linear classes (PY and IR) were also well distinguished.

However, the performance of different methods was not consistent. In contrast with Segnet and FCN-8s, we found that the Unet classification results contained a few undesirable noises, and PY and CL were confused with each other. GL was misclassified as DW and GH was misclassification as CD in the Unet classification results. Certain pixels in the linear class (PY) were also not clearly classified.

With respect to the results of Segnet and FCN-8s, all the misclassification problems produced by the Unet model were resolved with a higher accuracy. Thus, the incorrect classifications that appear in Figure 4 were revised accordingly.

## 3.2. Classification Accuracy Assessment

The classification performance of all methods was further investigated through quantitative assessment. Table 2 lists the classification accuracy assessment, including the overall accuracy (*OA*), *Kappa* coefficient (*Kappa*), and the class-wise precision (*PA*), recall (*RA*), and *F*1-score (*F*1).

**Table 2.** The classification accuracy among FCN-8s, Segnet, and Unet with three bands and four bands as input for the study area using the per-class precision (P), recall (R), *F*1-score (*F*1), overall accuracy (*OA*) and *Kappa* coefficient (*Kappa*). Per-class accuracy rates below 0.7 are colored pink-red, while accuracy rates over 0.9 are marked with bold. The overall highest *OA* and *Kappa* (last two rows) are marked bold. The method with the highest *OA* and *Kappa* is marked with a grey background.

| Class | 3 Band | | | | | | | | | 4 Band | | | | | | | | |
| --- | --- | --- | --- | --- | --- | --- | --- | --- | --- | --- | --- | --- | --- | --- | --- | --- | --- | --- |
| | FCN-8s | | | Segnet | | | Unet | | | FCN-8s | | | Segnet | | | Unet | | |
| | P | R | *F*1 | P | R | *F*1 | P | R | *F*1 | P | R | *F*1 | P | R | *F*1 | P | R | *F*1 |
| Cereal Land (CL) | 0.96 | 0.96 | 0.96 | 0.97 | 0.90 | 0.93 | 0.93 | 0.93 | 0.93 | 0.96 | 0.96 | 0.96 | 0.96 | 0.92 | 0.94 | 0.93 | 0.92 | 0.93 |
| Grassland (GL) | 0.91 | 0.93 | 0.92 | 0.84 | 0.91 | 0.88 | 0.85 | 0.85 | 0.85 | 0.90 | 0.94 | 0.92 | 0.94 | 0.86 | 0.90 | 0.82 | 0.91 | 0.86 |
| Poplar Land (PL) | 0.95 | 0.83 | 0.89 | 0.99 | 0.75 | 0.85 | 0.88 | 0.84 | 0.86 | 0.95 | 0.86 | 0.90 | 0.97 | 0.80 | 0.88 | 0.96 | 0.76 | 0.85 |
| Pathway in Farmland (PY) | 0.78 | 0.70 | 0.74 | 0.94 | 0.59 | 0.73 | 0.83 | 0.38 | 0.52 | 0.90 | 0.52 | 0.66 | 0.91 | 0.57 | 0.70 | 0.89 | 0.52 | 0.66 |
| Impervious Road (IR) | 0.78 | 0.80 | 0.79 | 0.93 | 0.78 | 0.85 | 0.85 | 0.77 | 0.81 | 0.79 | 0.79 | 0.79 | 0.97 | 0.66 | 0.78 | 0.90 | 0.83 | 0.86 |
| Sparse Woods (SW) | 0.96 | 0.92 | 0.94 | 0.96 | 0.77 | 0.86 | 0.81 | 0.74 | 0.77 | 0.97 | 0.92 | 0.95 | 0.97 | 0.79 | 0.87 | 0.88 | 0.74 | 0.80 |
| Dense Woods (DW) | 0.96 | 0.97 | 0.96 | 0.85 | 0.96 | 0.90 | 0.89 | 0.91 | 0.90 | 0.96 | 0.97 | 0.96 | 0.8 | 0.99 | 0.88 | 0.89 | 0.91 | 0.90 |
| Water Bodies (WB) | 0.89 | 0.97 | 0.93 | 0.98 | 0.88 | 0.92 | 0.95 | 0.96 | 0.95 | 0.94 | 0.97 | 0.96 | 0.97 | 0.89 | 0.93 | 0.97 | 0.93 | 0.95 |
| Waterless Channel (WC) | 0.93 | 0.93 | 0.93 | 0.97 | 0.85 | 0.91 | 0.85 | 0.89 | 0.87 | 0.91 | 0.94 | 0.92 | 0.97 | 0.84 | 0.90 | 0.91 | 0.88 | 0.89 |
| Water Conservancy Facilities (WF) | 0.94 | 0.91 | 0.93 | 0.96 | 0.91 | 0.93 | 0.98 | 0.90 | 0.94 | 0.91 | 0.97 | 0.94 | 0.96 | 0.86 | 0.91 | 0.95 | 0.95 | 0.95 |
| Construction Land (CD) | 0.95 | 0.94 | 0.95 | 0.91 | 0.95 | 0.93 | 0.89 | 0.95 | 0.92 | 0.94 | 0.96 | 0.95 | 0.96 | 0.94 | 0.95 | 0.91 | 0.96 | 0.93 |
| Greenhouse (GH) | 0.92 | 0.92 | 0.92 | 1.00 | 0.84 | 0.91 | 0.98 | 0.51 | 0.67 | 0.99 | 0.83 | 0.90 | 0.98 | 0.84 | 0.9 | 0.99 | 0.78 | 0.87 |
| Bare Land (BL) | 0.90 | 0.86 | 0.88 | 0.93 | 0.78 | 0.85 | 0.66 | 0.82 | 0.73 | 0.94 | 0.88 | 0.91 | 0.93 | 0.76 | 0.84 | 0.83 | 0.71 | 0.77 |
| *OA* | 93.9% | | | 89.9% | | | 87.9% | | | **94.2%** | | | 90.1% | | | 88.8% | | |
| *Kappa* | 0.92 | | | 0.86 | | | 0.85 | | | **0.93** | | | 0.87 | | | 0.86 | | |

From the table, when considering the *OA* and *Kappa* metrics, we found that the FCN-8s approach reported the best classification among Segnet and Unet. Averaged across the models, the *OA* and *Kappa* were 5% and 0.06 higher, respectively, in FCN-8s than the other two models. In the FCN-8s approach, the *OA* and *Kappa* were up to 94.2% and 0.93 in the four band input features, and 93.9% and 0.92 in three band input features, respectively. The four band input features for classification helped to promote accuracy more than three band input features, whether in FCN-8s, Segnet, or Unet, although the increased accuracies were slight. The increased accuracies in the *OA* and *Kappa* were 0.3% and 0.01 in the FCN-8s approach, 0.2% and 0.01 in the Segnet approach, and 0.9% and 0.01 in the Unet approach.

More specifically, considering the individual land class accuracy on the different experiments, the *PA*, *RA*, and *F*1 metrics performed differently. As illustrated by Table 2, CL, GL, DW, WB, WC, WF, and CD reported high *PA*, *RA*, and *F*1 metrics (in most cases > 0.9), while PL, SW, GH, and BL reported smaller ones. On the contrary, PY and IR resulted in lower accuracy rates (lower than 0.8), marked with pink-red in Table 2. The FCN-8s method showed the best performance for all classes, in particular for GL, PL, and SW, while the Segnet method helped to improve the *PA* metric in the PY, IR, WB, and WC classes. The Unet model had a good performance on the CL, WB, WF, and CD classes. Yet, adding the NIR band in classification did not show a significant increase in the accuracy of all classes. On the contrary, the *RA* metric in the PY class decreased 0.18 when using FCN-8s.

Figures 5 and 6 illustrate the normalized confusion matrices for the three methods with different input bands. As shown in Figures 5 and 6, there are differences in all classes among these methods. All the three methods accurately distinguished the CL, DW, CD classes with accuracies exceeding 90%, and yet the FCN-8s indicated slight improvements for GL, SW, WB, and WC compared to the Segnet and Unet methods, and the WF class performed worst in the Segnet approach. However, the class PY performed the worst in the classification methods. The best accuracy of PY was 0.7 in the FCN-8s approach with three band input features. This was possibly due to the lowest number of training samples for this class relative to the other land cover classes in this study.

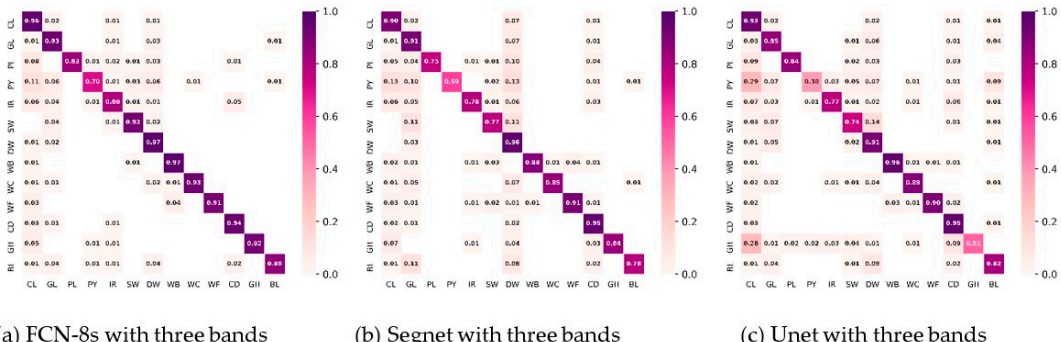

(a) FCN-8s with three bands　　　　(b) Segnet with three bands　　　　(c) Unet with three bands

**Figure 5.** Heatmap visualization of the normalized confusion matrix (normalized per row) for land cover classification using three bands input with three models. (**a**) FCN-8s with three bands, (**b**) Segnet with three bands, and (**c**) Unet with three bands. The labeled values on each grid show the normalized accuracy of each class, which indicates the degree of confusion ratio with the other classes. The normalized accuracies greater than or equal to 0.01 are labeled. CL represents cereal land, GL represents grassland, PL represents poplar land, PY represents pathway in farmland, IR represents the impervious road, SW represents sparse woods, DW represents dense woods, WB represents water bodies, WC represents the waterless channel, WF represents water conservancy facilities, CD represents construction land, GH represents greenhouse, and BL represents bare land.

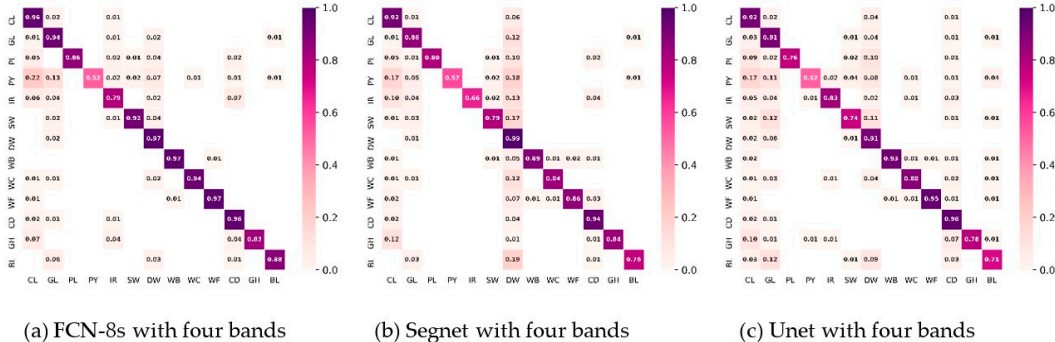

(a) FCN-8s with four bands     (b) Segnet with four bands     (c) Unet with four bands

**Figure 6.** Heatmap visualization of the normalized confusion matrix for land cover classification using four bands input with three models. (**a**) FCN-8s with four bands, (**b**) Segnet with four bands, and (**c**) Unet with four bands. The labeled values on each grid show the normalized accuracy of each class, which indicates the degree of confusion ratio with the other classes. The normalized accuracies greater than 0.01 are labeled. CL represents cereal land, GL represents grassland, PL represents poplar land, PY represents pathway in farmland, IR represents the impervious road, SW represents sparse woods, DW represents dense woods, WB represents water bodies, WC represents the waterless channel, WF represents water conservancy facilities, CD represents construction land, GH represents greenhouse, and BL represents bare land.

We found that PY was mainly confused with CL and secondly confused with GL and DW. We also found that adding the NIR band did not help to improve the accuracy of PY; on the contrary, it was slightly decreased. This is possibly because these PY and CL classes were adjacent successional, PY was also a linear shape, and the signatures in the NIR and R band were difficult to discriminate.

## 4. Discussion

In general, the FCN methods performed excellently in land cover classification. This is consistent with many results by researchers in the mapping field [7,25–29]. Spatial features in remotely sensed data are very important in classification and are intrinsically local and spatially invariant. CNN models the way that the human visual cortex works and enforces weight sharing with translation invariance, which enables the extraction of high-level spatial features from image patches.

To select a good FCN architecture in classification with high-spatial-resolution imageries is a difficult task for users [28]. This study showed that FCN-8s was a good choice in classification with high-spatial-resolution imagery. Although the FCN-8s, Segnet, and Unet are all pixel-based convolutional methods and showed state-of-the-art performance, they exhibited different results in their mapping accuracies. The FCN-8s model performed the best among these three architectures, and Unet showed the lowest accuracy of these three. The largest difference between these architectures was in the decoder stage. The upsampling method in the decoder stage has an important effect on the classification accuracy. In contrast with FCN-8s and Segnet, the Unet model also constructs a large number of feature channels in the upsampling stage by concatenating the channels from the decode layers, which allows the network to propagate the context information to higher resolution layers, which may lead to more training samples. The FCN-8s model combined three scale features in the decoder stage (seen in Figure 2), which may help to promote the accuracy with the fusion of different scales of spatial context information.

High-spatial-resolution remote sensing imagery with RGB+NIR bands performed better at mapping land cover than RGB bands only. As the NIR band contains more information regarding ground objects especially for vegetation types, adding this information helps to promote accuracy. Yet, adding the NIR band in FCNs did not demonstrate significant improvement in the overall accuracy. The most likely reason was that only one NIR band cannot provide enough information to improve the

classification. This research is consistent with the results of other researchers in comparing conventional machine learning methods and deep learning methods [15,16].

To train the RGB+NIR inputs, the network model requires very large training datasets. Thus, it is difficult to build a pre-trained network such as the one with the RGB input only. The number of classes in mapping is another important factor that will affect the accuracy. The majority of researches only focused on small class numbers in classification, such as five annotated land cover classes: built-up, farmland, forest, meadow, and waters in land cover classification [5]. However, when the number of classes is greater than 10, the overall accuracy decreased. We classified 13 classes in our paper and found the same phenomenon in our results: the accuracy of some classes were lower than other classes. Although the FCN-8s performed better than other methods, it was limited in the classification of certain classes. The most common reasons are that (1) the training samples were few, and (2) the classes shared similar spatial/spectral characteristics in RGB+NIR spectrum.

Different class types demonstrated different difficulty levels in the classification process. The linear-shaped classes showed a lower accuracy compared to other classes when using the FCN methods. This is likely because certain edge information of the linear object is lost in the convolution process. Therefore, a method for the linear object classification method should be considered in the future.

From this study, we found that GF2 imagery provided an encouraging result in estimating land cover based on the FCN-8s method. These results indicated that GF2 provides a comprehensive and large database of very high-resolution satellite images at an affordable cost, which can be exploited for large scale and land cover mapping.

## 5. Conclusions

Land cover mapping is an important terrestrial ecosystem variable that is used for many purposes. Using high-spatial-resolution images with multispectral information to map land cover is a very difficult task.

In this study, we adapted FCNs to classify high-spatial-resolution multispectral remote sensing imagery and quantitively compared the accuracy of different FCNs in classification with and without the NIR band as input. We gained certain insights through this process.

First, the FCN methods provided good results in land cover classification with high-spatial-resolution imagery. However, different FCNs architectures exhibited different results in their mapping accuracy. The FCN-8s model performed the best among these three architectures due to concatenating different spatial scale features in the upsampling stage.

Secondly, high-spatial-resolution remote sensing imagery with RGB+NIR bands performed well at mapping land cover compared with RGB bands only; however, the advantage was limited.

Thirdly, GF2 imagery provided an encouraging result in estimating land cover based on the FCN-8s method, which can be exploited for large-scale and land cover mapping.

**Author Contributions:** Conceptualization, Chonghuai Yao, Yuanyong Dian; data curation, Jingjing Zhou; formal analysis, Zemin Han, Hao Xia and Yongfeng Jian; funding acquisition, Yuanyong Dian; investigation, Zemin Han, Xiong Wang, and Yuan Li; methodology, Zemin Han; resources, Jingjing Zhou; supervision, Yuanyong Dian and Chonghuai Yao; writing the original draft, Yuanyong Dian; writing—review and editing, Hao Xia. All authors have read and agreed to the published version of the manuscript.

**Funding:** This research was supported by the National Key Research and Development Program (Grant No. 2017YFC0821900) and National Natural Science Foundation of China (Grant No.51778263).

**Acknowledgments:** The authors would also like to thank the China Centre for Resources Satellite Data and Application for providing the Gaofen-2 imagery.

**Conflicts of Interest:** The authors declare no conflict of interest.

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
