# Peer review of "Comparing Fully Deep Convolutional Neural Networks for Land Cover Classification with High-Spatial-Resolution Gaofen-2 Images"

_ijgi, doi:10.3390/ijgi9080478_

Round 1

Reviewer 1 Report

The title of this manuscript is really promising and the article could be interesting especially for chinese community but finally I found the paper poor in the content and in the structure. Which is the aim of this work? Where do you discuss it? In general,  by the reading, this article seems hastily written.

First of all, please, read carefully the MDPI guidelines for authors:  figures (such as 1 and 2 are incorrect; figure 1 is in Chinese and at low resolution; figure 2, 4 and 5 are unreadable ); captions are not homogeneous and not well explained (such as caption of figure 1); Table 2 is not according to the guidelines; etc.

There are many typos (e.g. lines 47, 60, 97, 128, 223, etc.).

Some  doubts:

Line 117: what are S1 and S2?

Line 118-120: this is a repetition of lines 95-100.

Lines 106-107: what does it means the 3 point? Where is the verb ?

Define if to use landcover or land-cover in the paper;

Finally, I suggest authors to give the paper a new dress in order  to make the work more sound because results could be interesting. In particular, please, rewrite carefully introduction, data info and abstract sections. 

In this form, this manuscript can not be published.

Author Response

1 ) The title of this manuscript is really promising and the article could be interesting especially for Chinese community but finally I found the paper poor in the content and in the structure. Which is the aim of this work? Where do you discuss it? In general,  by the reading, this article seems hastily written.

Response: Thanks very much for your comments. We carefully checked our paper and emphasized the aim of the paper. This study mainly focused on a systematic quantitative comparison different fully convolutional neural network (FCN-8s, Segnet, and Unet ) on Gaofen-2 imagery. We aimed to 1) evaluate the potential of using full convolutional neural network (FCNs) on high spatial resolution multispectral GF2 imagery in land cover classify; 2) conduct a systematic quantitative comparison different FCNs; 3) compare the classification accuracy when feeding RGB and RGB+NIR as input for GF2 satellite imagery. The objects were rewritten in section 1 line 139. In the discussion section, we discussed the reason for the different performances of FCNs. And, we discussed the reasons why different inputs will cause different accuracy. Furthermore, the effects of class number and class types in overall accuracy were also discussed.

2 ) First of all, please, read carefully the MDPI guidelines for authors:  figures (such as 1 and 2 are incorrect; figure 1 is in Chinese and at low resolution; figure 2, 4 and 5 are unreadable ); captions are not homogeneous and not well explained (such as caption of figure 1); Table 2 is not according to the guidelines; etc.

Response: Thanks very much for your comments. We modified all the captions of figures and tables. The detailed information introduced in the captions. We also redrewn the figures to derive 600 DPI figures.

3) There are many typos (e.g. lines 47, 60, 97, 128, 223, etc.).

Response: Thanks very much for your comments. We have revised the whole manuscript carefully and tried to avoid any grammar or syntax errors.

4) Line 117: what are S1 and S2?

Response: Thanks very much for your comments. Figure 1 was redrawn and added the label of different scenes and sample areas.

5) Line 118-120: this is a repetition of lines 95-100.

Response: Thanks very much for your comments. We agreed with your comments. The repetitions were deleted.

6) Lines 106-107: what does it means the 3 point? Where is the verb ?

Response: Thanks very much for your comments. We rewrote this sentence and add a verb “compare” in this sentence.

7) Define if to use landcover or land-cover in the paper;

Response: Thanks very much for your comments. All the terms “landcover” or “land-cover” in the paper were changed to “land cover”.

8) Finally, I suggest authors to give the paper a new dress in order  to make the work more sound because results could be interesting. In particular, please, rewrite carefully introduction, data info and abstract sections.

Response: Thanks very much for your comments. We carefully checked our paper and reorganized the paper especially in abstract and introduction sections.

In the abstract section, the detailed results were added in and the aims of our paper emphasized.

In the introduction section, we reorganized the logical order of the article. First, we emphasized the importance of the land cover; second, we introduced the land cover products with remote sensing imagery, and then, the limits of traditional land cover products were depicted. Third, the challenge and methods of land cover classification with high spatial resolution imagery were introduced and then the deep learning method and its problems were listed. At last, we showed our objects in this study.

In the materials and methods section, we redrew the figures and described the dataset.

Reviewer 2 Report

The authors present a comparison between different deep convolution Neural network approaches for land cover classification using Gaofen – Images.

Their review of existing work is satisfactory. They clearly explain the methodology that are using for the comparison the selected methods the study area and their evaluation methodology. They also clearly present their experimental results, while their conclusions are supported by their results.

The manuscript needs extensive English writing check.

Some examples

In line 73 development should change to developed

In line 88 comm ones should change to the common ones

In line 126 water bodies are written 2 times

Line 132-133 as the same should change at the same

And so on

Author Response

1) The manuscript needs extensive English writing check. Some examples In line 73 development should change to developed; In line 88 comm ones should change to the common ones; In line 126 water bodies are written 2 times; Line 132-133 as the same should change at the same and so on.

Response: Thanks very much for your comments. We have carefully checked the paper and modified errors.

Reviewer 3 Report

Quality of the figure 1 and 2 should be improved.

Author Response

Response: Thanks very much for your comments. We have modified the all the figures. And upload the original resolution picture to the website.

Round 2

Reviewer 1 Report

The manuscript has been improved and after minor corrections it is ready to be published.

  • Please, change "Fig.nn" in "Figure nn" in the text when you mention figures.
  • Figures 2, 5, and 6 continue to be unreadable. Please, look for a way to change them.

Author Response

Please, change "Fig.nn" in "Figure nn" in the text when you mention figures.

Thanks very much for your comments. We modified all these text in paper.

Figures 2, 5, and 6 continue to be unreadable. Please, look for a way to change them.

Thanks very much for your comments. We redraw the Figure 2 and add some descriptions on the title of this page. We added som descriptions on the Figure 5 and 6.